# Unsupervised Outlier Detection in IOT Using Deep VAE

**DOI:** 10.3390/s22176617

**Published:** 2022-09-01

**Authors:** Walaa Gouda, Sidra Tahir, Saad Alanazi, Maram Almufareh, Ghadah Alwakid

**Affiliations:** 1Department of Computer Engineering and Network, College of Computer and Information Sciences, Jouf University, Sakaka 72341, Al Jouf, Saudi Arabia; 2Electrical Engineering Department, Faculty of Engineering at Shoubra, Benha University, Cairo 13518, Egypt; 3University Institute of Information Technology, PMAS Arid Agricultural University, Rawalpindi 46000, Pakistan; 4Department of Computer Science, College of Computer and Information Sciences, Jouf University, Sakaka 72341, Al Jouf, Saudi Arabia; 5Department of Information Systems, College of Computer and Information Sciences, Jouf University, Sakaka 72341, Al Jouf, Saudi Arabia

**Keywords:** unsupervised, IoT, outliers, variational auto-encoder, deep learning

## Abstract

The Internet of Things (IoT) refers to a system of interconnected, internet-connected devices and sensors that allows the collection and dissemination of data. The data provided by these sensors may include outliers or exhibit anomalous behavior as a result of attack activities or device failure, for example. However, the majority of existing outlier detection algorithms rely on labeled data, which is frequently hard to obtain in the IoT domain. More crucially, the IoT’s data volume is continually increasing, necessitating the requirement for predicting and identifying the classes of future data. In this study, we propose an unsupervised technique based on a deep Variational Auto-Encoder (VAE) to detect outliers in IoT data by leveraging the characteristic of the reconstruction ability and the low-dimensional representation of the input data’s latent variables of the VAE. First, the input data are standardized. Then, we employ the VAE to find a reconstructed output representation from the low-dimensional representation of the latent variables of the input data. Finally, the reconstruction error between the original observation and the reconstructed one is used as an outlier score. Our model was trained only using normal data with no labels in an unsupervised manner and evaluated using Statlog (Landsat Satellite) dataset. The unsupervised model achieved promising and comparable results with the state-of-the-art outlier detection schemes with a precision of ≈90% and an F1 score of 79%.

## 1. Introduction

IoTs are virtualized, integrated, and networked representations of objects in our environment. The integration of smart home devices, daily gadgets, health sector, and business are managed cloud infrastructure networks that are not bound by administrative, geographical, or cultural boundaries. Different sensors, instruments, actuators, and a variety of other devices and satellites enhance IoT-based applications. They are critical in gathering and comprehending the data supplied by smart city devices, agriculture, manufacturing, network security, and healthcare. This dataset collection is used to suggest and enhance services, security, and public utilities [1]. Wearable gadgets, distributed sensor-based smart devices, and smart devices, in general, may communicate information and services through a network, which is the basis of smart technology [2].

Outlier detection is another name for anomaly detection. Novelty discovery is the task of finding “odd” samples from a large dataset. Anomaly detection has grown in importance in enterprises where damaged goods or failures may be regarded as abnormal samples [3]. However, due to the rapid growth of big data technology and the Internet of Things (IoT), more research is being conducted in this area. Anomaly detection approaches have also received considerable interest in healthcare systems, smart city surveillance, and satellite-related information systems [4].

Despite their vast applications in various domains, there are several issues associated with IoTs and the data generated by IoT. First of all, different IoT-based solutions generate massive amounts of complex dimensional data, prompting the use of more advanced intelligence algorithms to evaluate and uncover hidden complex patterns in the data [5]. Second, conventional outlier detection techniques are frequently problem-specific and have supervised procedures [6]. The system based on these techniques needs domain knowledge and extensive data labels. As a result, in data-driven IoT systems, designing an unsupervised outlier detection approach is vital for smart and digital cities [7]. Among these issues outlier detection is a critical issue because of these challenges. Within that high-dimensional data, conventional anomaly detection techniques also fail to detect and classify anomalies [8].

Determined by the presence of labels in the dataset, outlier detection can be classified into three types, namely supervised, semi-supervised and unsupervised outlier detection techniques. Using properly labeled training data as well as test data, in training data, supervised outlier detection seeks to distinguish between normal and abnormal occurrences [9]. The model will then be implemented in both the training data and test data. A model using just normal examples for semi-supervised abnormality detection trains to create a standard “normal” model, allowing normal features to be learned equitably [10]. Outliers are defined as data that deviates significantly from the “normal” exemplary, as intended by scoring high. The unsupervised abnormality detection method is based on unlabeled data and does not differentiate between training and test data [11,12]. It makes a significant assumption, for example, that the detected data consist of a small number of abnormalities and a large number of typical cases. Models can be used to learn data’s intrinsic information. Outliers will be captured if there are any variances from the significant cases.

Unsupervised abnormality detection is considered the most challenging task among the three classes due to data variety and information scarcity. Furthermore, labeling is a costly and time-consuming needed task in real-world applications [13,14,15]. As a result, the lack of labels is common in many situations, making unsupervised abnormality detection techniques very useful and suitable.

A Variational Auto-Encoder (VAE) is a graphical model based on directed notes with a probabilistic approach, and its posterior is estimated by a neural network, resulting in an architecture similar to Auto-Encoder [16,17,18]. The encoder–decoder structure of the Variational Auto-Encoder allows for learning and training a mapping from highly complex and structured dimensional input and transforms to a latent representation having low dimensions. This is achieved while maintaining reconstruction accuracy at a higher level [19]. Furthermore, our strategy optimizes the performance of cooperative cooperation by combining feature extraction and model creation. The encoder–decoder structure uses a Fully Convolutional Network (FCN) without a completely connected layer. This layer retains relative spatial coordinates between the input dataset and the output features. This type of DNN is called a Fully Convolutional Variational Auto-Encoder (FC-VAE) [20]. The unsupervised abnormality detection challenges will be the focus of this research. Whether labels are provided or not, the anomaly detection problem may be viewed as a special unbalanced classification problem. It is well acknowledged that data quality has a significant impact on the outcomes of data-driven strategies.

This article proposes a deep learning system that is end-to-end and only training on unprocessed data using VAE. Our solution is based on the premise that normal data can be linked to a Variational Auto-Encoder. Then an outlier is defined as a test data instance that is not possible to link to any VAE component. Different varieties of features generally translate to better outcomes. However, most of the time, due to the presence of irrelevant and duplicated data, the quality of data cannot be certain. The worthless information not only reduces the accuracy of outlier detection but also lengthens processing complexity, and more time is consumed [3].

Our approach is based on the Variational Auto-Encoder, a probabilistic clustering model. The following are the research’s standout contributions.

We devised an unsupervised VAE that can detect abnormality effectively on satellite data.We used the VAE from its latent representation to detect correlations between satellite data and identify anomalies.We conducted an experimental evaluation to show that the proposed VAE outperforms cutting-edge approaches.

The remaining sections of the paper are organized as follows. Section 2 presents a summary of existing relevant research efforts. Section 3 describes the technique used to gather and analyze the satellite information, and Section 4 proposes a VAE-based system. Section 5 discusses the experiment and the outcomes, which are presented in detail. Section 6 concludes with a conclusion and possible research prospects.

## 2. Related Work

In the following part, we reviewed studies in the realm of deep learning that used VAE to discover abnormalities or outliers. For decades, researchers have been researching outlier detection techniques. Depending on whether labels are employed during training, existing research may be classified as supervised, semi-supervised, or unsupervised outlier detection.

### 2.1. Deep Learning (DL)

Deep Neural Networks have captured a major area of research in the domain of machine learning in the past decade [21,22,23,24]. The result of this effort has proved extraordinary results across a wide series of application fields. Taking DL as a subtype of machine learning (ML) that learns to represent data as a layered hierarchy of concepts within neural network layers to attain high performance and flexibility. As the size of the data rises, deep learning beats regular machine learning, as seen in Figure 1. Anomaly detection algorithms based on DL have grown in reputation in recent years and are applied to various applications; studies have shown that deep learning totally outperforms previous approaches [25,26].

DL is highly recommended for anomaly detection and producing cutting-edge results, as shown in Figure 1. Ravanbakhsh [27] was among the first to apply deep learning to anomaly detection through the use of a flow of auto-encoders (AE). It made use of both the auto-reconstruction encoder’s error and a sparse auto-encoder sparsity assessment. Ravanbakhsh [27] explored a Fully Convolutional Network (FCN). This FCN was needed as a pre-trained model. It helped in an effective quantization double layer. There was a net’s final layer in the model to identify progressive CNN patterns. They demonstrated a novel method for detecting local abnormalities by merging progressive CNN patterns with a hand-crafted attribute. Ravanbakhsh [27] extracted discriminative video unique properties using fully convolutional neural networks (FCNs). A normal event was represented by a Gaussian distribution, while a test region that differed from the normal reference model was labeled as an anomaly. In another work, Hasan [28] explored temporal regularity while using a fully linked AE as well as a Fully Convolutional AE. However, decisions were made based on local features and brief video clips. The reconstruction errors were transformed into a regularity score to find the anomalies.

In order to consider anomaly detection as a classification problem, Guo designed an active learning-based supervised scheme [19]. Li used the supervised ID3 decision tree method to detect anomalies in computer networks [29]. Furthermore, Zaib demonstrates a model that is hybrid and used for malicious code detection [30]. It takes into account both AE and multilayer Restricted Boltzmann Machines (RBM) [31].

In the form of current multimedia, we present a hybrid deep-learning-based anomaly detection approach for suspicious flow identification [32]. It contains two units: (1) an anomaly detection unit that detects abnormal activities using an enhanced restricted Boltzmann machine and SVM based on gradient descent and (2) an end-to-end data delivery unit that meets the SDN’s stringent QoS requirements such as higher bandwidth and lower latency. Finally, the suggested technique was evaluated using real-time and benchmark datasets to establish its utility and performance in anomaly recognition and data transmission.

### 2.2. VAE along with DL Techniques

VAE is a novel and nonlinear feature extraction scheme. VAE can effectively and stably signify a data structure [23]. However, feature quality is a critical factor in conventional unsupervised outlier detection schemes. The focus of their research was to detect network traffic intrusions [10]. They used unsupervised DL methods and merged them with a learning approach that was semi-supervised. Flow features were used in VAE to detect and identify anomalies and intrusions. These Flow-based features were obtained from network traffic data. The sources also mentioned other forms of attacks that were employed in the studies. They examined the area under the ROC curve and Receiver Operating Characteristics (ROC), as well as the One-Class SVM. The resulting ROC curves were thoroughly reviewed in order to evaluate the approaches’ performance at various threshold levels.

In existing DL models, unsupervised learning techniques were designed for accurate COVID-19 prediction [33]. Their research proposed a novel unsupervised Deep Learning based VAE (UDL-VAE) model for the detection and classification of COVID-19. To increase picture quality, Wiener filtering (AWF), an adaptive-based preprocessing approach, was applied using UDL-VAE. For feature extractor Adagrad technique is practiced [34,35]. For classification, an unsupervised VAE model is utilized. A series of tests were carried out in order to emphasize the effective outcome of the UDL-VAE model and demonstrate its outstanding diagnostic performance. The experimental results proved the efficacy of the UDL-VAE model, with 0.98 and 0.92 accuracies on binary and multiple classes, respectively.

In their paper, they describe soft sensor modeling as a VAE-based just-in-time (JIT) learning system [36]. In industrial processes, JIT learning is frequently used for soft sensor modeling. Traditional JIT learning methods ignore the ambiguity in variables. The VAE methodology is modified to extract features from a noisy input data set in order to improve standard JIT learning approaches. A Gaussian distribution is used to express each feature variable. The Kullback–Leibler divergence (KLD) is then used to determine the degree of similarity between the query sample and historical samples [35,36]. Furthermore, during modeling, historical samples that are most equivalent to the query samples are chosen based on the KLD values. Finally, Gaussian process regression is employed as a nonlinear regression model to describe and forecast the relationship between the selected input samples and the matching output samples. The proposed method’s effectiveness was validated by an arithmetical example and its application to a real-world debutanizer industrial process. Table 1 shows different methods for anomaly detection that are based on DL and used AE or hybrid AE schemes to determine anomalies or outliers in their dataset.

## 3. Materials and Methods

In this section, we present the dataset description used in this research work. Once dataset is acquired, preprocessing is conducted on a tabular dataset of Satellite (Landsat Satellite). Then, the VAE is described in detail to show its working mechanism besides its importance in our work. After that, the proposed framework is presented that is based on VAE. In this study, we used a Satellite input database to train a VAE model.

### 3.1. Dataset Description

Data are central to deep learning models, and they serve as fuel for these learning models. We gathered satellite datasets in tabular form for this project from a publicly accessible data source. Statlog (Landsat Satellite) is a multi-class classification dataset from the UCI machine learning repository. The Statlog (Landsat Satellite) contains a total of 6435 samples, where the total normal samples = 4399 and the total outlier samples = 2036. These samples are stored in tabular form and have 36 features or dimensions.

### 3.2. Dataset Pre-Processing

The dataset of Satellite (Landsat Satellite) was available in tabular form. We standardize features by removing the mean and scaling to unit variance. This process is considered a pre-processing step in the proposed framework.

### 3.3. Variational Auto-Encoder Algorithm

A variational autoencoder (VAE) is a type of generative model in which the distribution of the encodings is regularized during training. In other words, VAE gives the latent space structure. The internal structure of the multidimensional latent space for a well-learned model defines its properties. The decoder component reconstructs the input using this information. The VAE architecture is made up of an encoder and a decoder with an intermediate compressed low-dimensional layer space, just as the AE architecture. The encoder maps the data to a posterior distribution, which is exclusive to VAE. The use of the univariate Gaussian distribution is a popular option in VAE. The Kullback–Leibler divergence measure between the approximation output and the target (input) characterizes the regularized term. The sum of the reconstruction loss and the Kullback–Leibler divergence is the overall loss in VAE, as shown in Algorithm 1.

A VAE has been shown to be successful in outlier detection as well as providing a wide range of complicated data. This assumes that normal samples’ latent representation is compatible with a Gaussian distribution. It means that all training datasets are grouped in feature space, and anomalies are located distant from the cluster [47]. Through a latent space, VAE can acquire the feature distribution for high-dimensional input data.

This study employs a fully connected neural network in both the encoder and decoder levels [48].


**Algorithm 1** VAE base outlier detector
**Input**
Normal dataset D
Dataset with abnormal d^(i)^ = {1,2,3……N}
A thresh hold value α
**Output**
Reconstruction probability class labels
**Steps**
ФѲ ← train using VAE with normal dataset D
**   **
**1.**
For loop from k = 1 to N
**   2.**
Calculate       $\mathrm{Reconstruction}\;\mathrm{error}\;\left(k\right)=\|d^{k}-g\theta \left(f\emptyset \left(d^{k}\right))\right\|$

**   3.**
   if Reconstruction error (k) > α then
                   outlier ← d^(k)^
          Else
                   Not outlier ← d^(k)^
          end if
          End for


Where gθ(f∅ (d^(k)^) is a recognition network also known as probabilistic encoder. f∅(d^(k)^) is a generative decoder as a generative network.

The VAE system provided here extracts useful characteristics for numerous standard unsupervised outlier detection applications. The reconstruction probability is computed using the stochastic latent variables that create the parameters from the distribution of the original input variable. The constraints of the input variable distribution, not the input variable itself, are reconstructed. This is the likelihood of data being generated from a particular latent variable pulled from the approximate posterior distribution.

### 3.4. Proposed Framework

This section explains an unsupervised-technique-based framework that relies on a deep Variational Auto-Encoder (VAE) to detect outliers in Satellite data by leveraging the characteristic of the reconstruction ability as well as VAE’s low-dimensional representation of the input data’s latent variables. Figure 2 demonstrates the DL approach that uses Landset Statlog (Landsat Satellite) to learn the candidate and favorable features to differentiate between normal and outlier samples. As in Algorithm 1, Standard Scaler is executed as a preprocessing phase. Then, in the next phase, A VAE is used to obtain the structure of the latent space. The VAE architecture, such as AE, is made up of two primary parts: a decoder and an encoder with an intermediate compressed low-dimensional layer space. The internal structure of a well-learned model defines the multidimensional latent space. The decoder section uses this information to rebuild the input. The sum of the reconstruction loss between the input and the reconstructed output and the Kullback–Leibler divergence is then used as the total loss in VAE, which in turn is used as an outlier score for our data.

VAE is used to achieve these goals since it is capable of coping with data uncertainties and collinearity in the data set [49]. As previously explained, VAE is an unsupervised neural network whose purpose in its output layer is to generate/reconstruct the provided input samples [50]. Effective features for representing the high-dimensional, redundant, and noise-corrupted input data are generated by training a VAE model on input data. We may categorize the samples as normal or outlier based on the mean squared error between the original observation and the reconstructed one. During the optimization phase, a classifier’s threshold value is calculated.

## 4. Experimental Setup

Several tests were carried out using the Statlog (Landsat Satellite) dataset to illustrate the efficacy of the proposed VAE-based anomaly detection approach. These tests also yielded unique results, which were utilized to compare the suggested model to state-of-the-art techniques. The code for the proposed system was developed using TensorFlow -Keras libraries in Python Jupyter Notebook on Ubuntu 20.04.4 desktop pc with Core i7-11700F CPU with 16 GB RAM and NIVIDIA GeForce RTX 3060 GPU with 12 GB Memory. During the experiments, the training-to-testing ratio was 80:20%. According to the suggested training scheme, training was performed utilizing 80 percent random samples of Statlog (Landsat Satellite) Data Set for the proposed DL-based VAE system.

Only normal samples were used to train the model, so outlier samples from the training set were excluded and added to the test set. Hence, the model was tested using all 2036 outlier samples besides the test normal samples. The outlier score was generated by calculating the reconstruction error between the input and the reconstructed sample. The suggested framework is pre-trained on Statlog (Landsat Satellite) Data Set using the Adam with a learning rate technique that reduces the learning rate when learning gets stagnant for a period of time (i.e., validation patience). The Adam optimizer was trained with the following hyper-parameters: Total number of epochs = 200; batch size = 16, learning rate = 1 × 10^−3^, and loss function = “elbo”.

### Assessment Methods

To assess performance, we used the performance indicators stated in Equations (1)–(6) to compare our proposed system to other systems. The proposed evaluation parameters were taken from existing research work [51,52].
(1)$$TRP=\frac{TP}{\left(TP+FN\right)}$$
(2)$$TRN=\frac{TN}{\left(TN+FN\right)}$$
(3)$$PPV=\frac{TP}{\left(TP+FP\right)}$$
(4)$$FNR=\frac{FN}{\left(TP+FN\right)}$$
(5)$$FPR=\frac{FP}{\left(FP+TN\right)}$$
(6)$$\;\;\;\;\;\;\;\;\;\;\;F1-Score=\frac{2*\left(Precision*recall\right)}{\left(Precision+recall\right)}$$

True positives (samples correctly recognized as having outlier), true negatives (samples correctly identified as not having outlier), false positives (samples with outliers other than anomaly), and false negatives (samples with anomaly identified as normal samples).

## 5. Discussion

The results of the suggested systems’ different trials utilizing the Statlog (Landsat Satellite) dataset are shown in this section. The outcomes defined the dataset into two categories: outliers and normal instances. Figure 3 depicts the frequency of normal instances of satellite images, as well as the frequency of outliers. From the given dataset, there are 4399 normal samples and 2036 outlier samples.

The effectiveness of the proposed technique is assessed using ROC and AUC assessments. The AUC measure is used for assessment since it is the benchmark in unsupervised outlier identification, as well as its interpretability. The ROC-based evaluation was used since it was insensitive to class distribution. It also indicates that one technique may outperform another at different thresholds. Furthermore, the Mean Precision–Recall (MPR) curve is used as a statistic to assess and compare the efficacy of outlier identification systems.

We also demonstrate reconstruction probabilities, when employed as the outlier score, may discriminate between the abnormal and the normal. Figure 4 displays the distribution of the obtained reconstruction error for both normal and anomalous samples. To properly reflect the distribution, we conducted operations on each sample by obtaining the Negative Logarithm of the Reconstruction Probability (NLRP). The likelihood of each sample being anomalous grows proportionately to the value of NLRP. According to Figure 4, the majority of normal samples have lower NLRP values than anomalous samples for the used dataset. The distributions of reconstruction probabilities from normal and anomalous samples are then easily differentiated with very minimal overlaps.

The entire tradeoff between the true (genuine) positive rate and the false positive rate is generated using Receiver Operating Characteristic (ROC) curves. The area under the ROC curve can be used to summarize the ROC curve. This is referred to as the AUC. The AUC ROC curve is a graph that shows the classification performance of a model in terms of two variables: true positive (tp) and false positive rate (fpr). The area under the receiver operating characteristic (AUROC) and Mean Precision–Recall (MPR) are used as metrics throughout the assessment to evaluate and examine the effectiveness of outlier identification systems. An effective model must have a high AUROC, but AUPRC frequently exposes the gap between techniques when dealing with imbalanced datasets. The suggested FC-VAE framework’s efficiency was assessed using AUC. The AUC may be determined by incorporating the areas of small geometrical components under the ROC curve. Figure 5 displays the AUC assessments of the suggested architecture. Existing classification models are outperformed by the proposed model (AUC = 0.842).

Figure 6 shows the confusion matrix of the outlier and normal sample from the Statlog (Landsat Satellite) dataset. According to the graph, 743 samples were correctly classified as normal samples, 164 normal were misclassified as abnormal samples, and 597 outlier samples were incorrectly classified as normal samples. The Satellite dataset contains 1439 outliers that are correctly classified as anomalies.

This section shows the performance of the proposed Variational Auto-Encoder model using the MPR evaluation method and summarizes the results using the accuracy and recall curves. This curve focuses mostly on the performance in the abnormality class and is thus significantly more indicative when the abnormalities are only of interest to us. The Mean Precision–Recall (MPR) curve is used as a measure during the assessment to evaluate and compare the performance of abnormality detection systems. MPR summarizes the precision and recall curve, which focuses on performance in the abnormality class and is therefore much more revealing when the abnormalities are just of interest. MPR is calculated with the help of a confusion matrix using Equations (1)–(6). Precision or positive predictive value (PPV) = 0.897 is recorded, whereas recall is calculated as Recall (TPR) = 0.70; hence our model has achieved a higher PPV of ≈90%, as shown in Figure 7.

While inspecting our system results, the obtained F1 score is 0.79. This s crucial for a practical outlier detection system. Moreover, the proposed system shows that the FPR = 0.18 and specificity (TNR) appears to be 0.81, and the false negative ratio is FNR = 0.29 with AUC_ROC = 0.842. These results are then compared with state-of-the-art approaches used in the same domain. As shown in Table 2, FC-VAE gives remarkable results in terms of precision and accuracy like existing approaches. When compared with other state-of-the-art approaches that utilize satellite dataset (images or tabular form), different results are concluded in terms of precision, recall, F1 score, and accuracy. Most of the recall value lies in the range of 85–100%. The precision varies in the range of 78–92%. The accuracy of most of the techniques achieved during the evaluation reached 98%. F1 score, which is the harmonic mean of precision and recall, varied from 83% to 94%.

It is vital to note that comparing outcomes in this area of unsupervised learning is highly challenging owing to the wide range of accessible datasets and techniques. Furthermore, the clustering labels may seem different, and the performance indicators may vary. As certain dataset labels are never completely recognized, it may result in presenting findings in ambiguous test datasets. It is evident from this that the performance of the predicted FC-VAE may change if the prediction is based on a different set of training or test datasets.

The proposed FC-VAE model was tested on the KPI time series dataset used for VAE-GEN [54]. We made a public request to the researchers to reveal the dataset’s specifics. The authors were gracious enough to publish their model’s source code as well as the dataset on Jupiter Notebook. We executed FC-VAE model on their dataset and attained the acclaimed accuracy of 76% [54]. Table 3 compares the accuracy acquired as a consequence of this comparison experiment with the Statlog dataset used for FC-VAE. It is incredible to note that the satellite dataset provided considerably better findings in terms of precision and F1 score, indicating the generality of the proposed approach.

## 6. Conclusions

IoT-based applications are becoming prevalent in a variety of fields. IoT data may contain outliers or display aberrant behavior as a consequence of attack activity or device failure. The bulk of current outlier identification methods relies on labeled data, which is typically difficult to obtain in the IoT sector. In this paper, an unsupervised approach based on deep Variational Auto-Encoder (VAE) is presented. The purpose was to find outliers in IoT data by exploiting the VAE’s reconstruction capabilities and low-dimensional representation of the input data’s latent variables. The proposed model was trained in an unsupervised fashion using just normal data with no labels and assessed using the Statlog (Landsat Satellite) dataset. The unsupervised model produced promising and comparable results to state-of-the-art outlier identification approaches, with an accuracy of 90% and an F1 score of 79%. With such promising findings, we intend to expand our technique to utilize a better strategy employing AE architecture in order to manipulate the abnormality detector with datasets other than the ones it has been trained on, hence enhancing the performance and accuracy of unlabeled data.

## Figures and Tables

**Figure 1 sensors-22-06617-f001:**
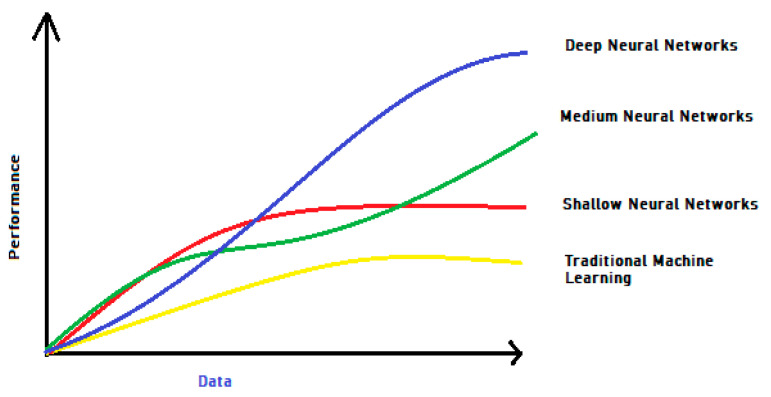
Traditional ML vs. DL algorithms performance assessment (Caroppo, Leone, and Siciliano, 2020).

**Figure 2 sensors-22-06617-f002:**
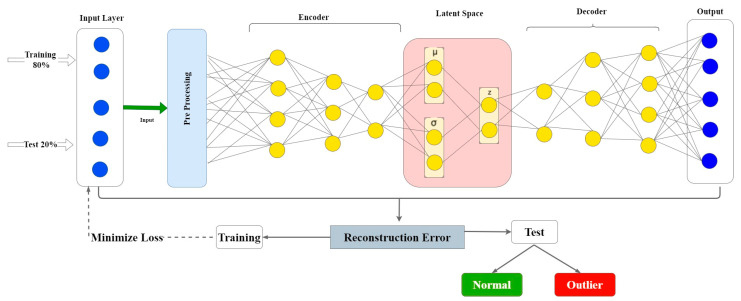
Proposed Variational Auto-Encoder framework for outlier detection.

**Figure 3 sensors-22-06617-f003:**
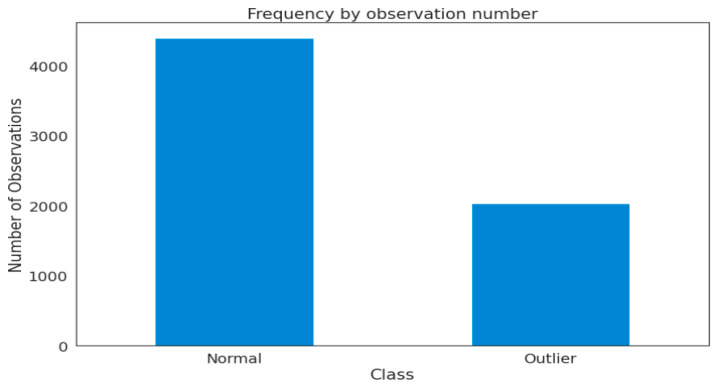
Visualization of the distribution of normal/outlier instances in the used dataset.

**Figure 4 sensors-22-06617-f004:**
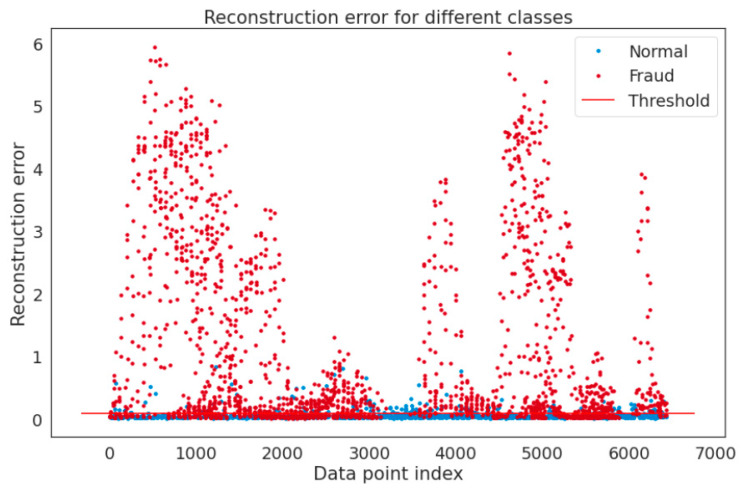
Different classes depicting reconstruction error.

**Figure 5 sensors-22-06617-f005:**
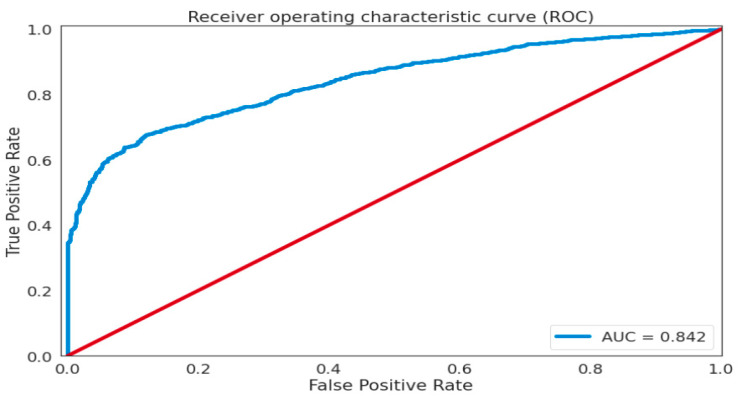
Receiver operating characteristic curve (ROC) for Statlog (Landsat Satellite) dataset.

**Figure 6 sensors-22-06617-f006:**
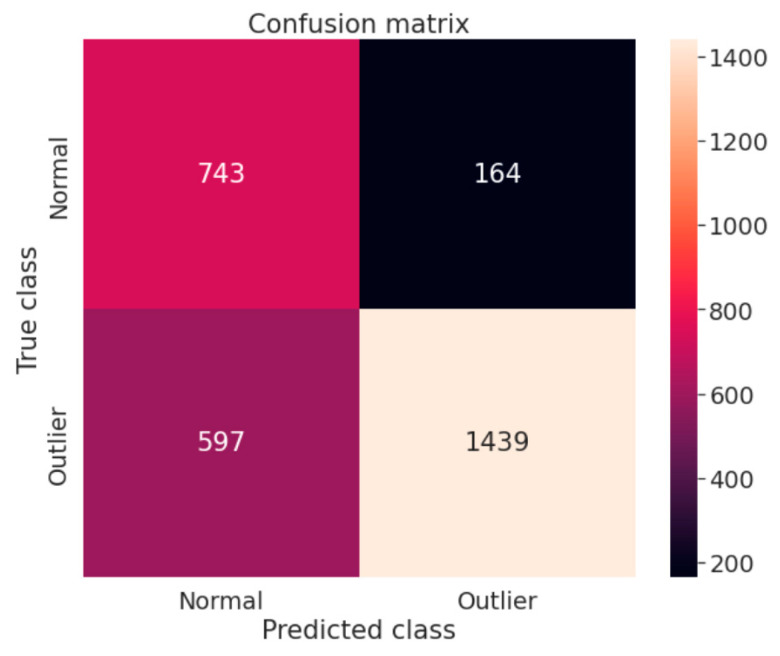
Confusion matrix with Statlog (Landsat Satellite) dataset.

**Figure 7 sensors-22-06617-f007:**
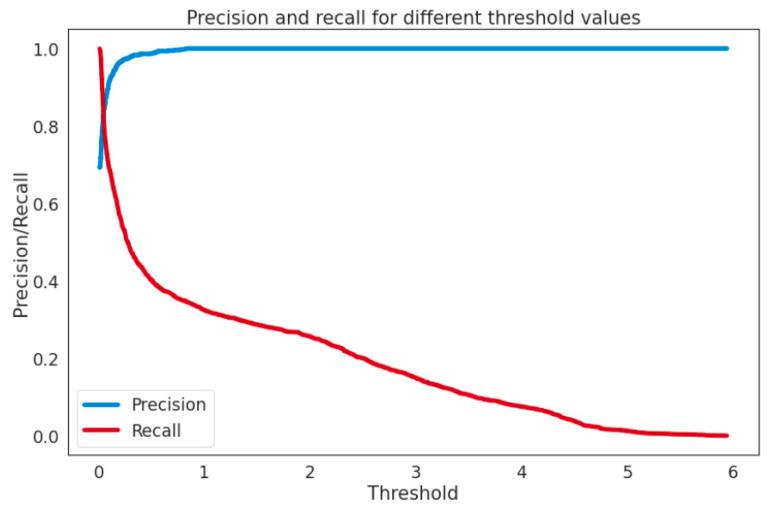
Precision and recall trend of Statlog (Landsat Satellite) dataset.

**Table 1 sensors-22-06617-t001:** DL-based anomaly detection techniques using AE or Hybrid AE.

Method	Ref.	Data	Objective	Architecture	SUP.
OADA	[37]	Video	Reconstruction	AE	Semi.
RandNet	[38]	Tabular	Reconstruction	AE	Unsup.
UODA	[39]	Sequence	Reconstruction	AE&RNN	Semi
GT	[40]	Image	Classification	CONV	Semi
E3 Outlier	[41]	Image	Classification	CONV	Semi
AE-1SVM	[42]	tabular and image	One-class	AE & CONV	Unsup.
DAGMM	[43]	Tabular	Clustering	AE & MPL	Unsup.
AEHE	[44]	Graph	Anomaly score	AE & MPL	Unsup.
ALOCC	[45]	Image	Anomaly score	AE & CNN	Semi
OCAN	[46]	Sequence	Anomaly score	LSTM-AE & MLP	Semi

**Table 2 sensors-22-06617-t002:** Comparison of proposed FC-VAE with state-of-the-art techniques.

Technique	Ref.	Method	Learning Technique	Results
MCD-BiLSTM-VAE.	[49]	Deep Learning	unSup	P 0.92, R 0.99, F1 0.94, Acc 0.98
GNN-DTAN	[50]	Neural Network	Unsup	R 0.85, P 0.82, F1 0.83, Acc 0.98
Pseudo-period technique		Machine Learning	Unsup	Acc 0.93
One class	[51]	Machine Learning	Sup	P 0.78, R 1.0, F1 0.87, AUC 0.95
EML	[52]	Machine Learning	Sup	F1 0.93, Acc 0.98
SLM	[53]	Machine Learning	Unsup	Performance 0.90
FC-VAE		Deep Learning	**Unsup**	**AUC ROC 0.842, P 0.897, R 0.706**

**Table 3 sensors-22-06617-t003:** Evaluation of proposed model with different datasets.

Technique	Dataset	Learning Technique	Results
VAE-GAN	Time series	UnSup	P 0.76, R 0.5, F1 0.6
FC-VAE	Statlog	UnSup	P 0.897, R 0.706, F1 0.79

## Data Availability

Not applicable.

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
