# Peer review of "Unsupervised Outlier Detection in IOT Using Deep VAE"

_sensors, 2022, doi:10.3390/s22176617_

Round 1
Reviewer 1 Report
In this paper, the authors describe the outliers detection in the Statlog (Landsat Satellite) dataset using a deep Variational Auto-Encoder (VAE). The purpose is to develop and test an unsupervised strategy to identify two classes of data: normal and outlier. This problem is interesting and essential for cleaning datasets before the classification task. The results are competitive. However, I consider that the proposed approach is not clearly described and compared with other strategies in the literature. I recommend considering the following concerns:
1) The abstract should contain less context on IoT and include this proposed method’s novelty and advantages.
2) Pictures of figure 2 show low resolution; this diagram should include the number of classes and more detail about the use of the dataset (80%-20%)
3) Deep VAE is theoretically few described; subsection 2.3 should be in section 3. Moreover, section 3 is confused; please check subsections 3.2 and 3.3.
4) Experimental results are also confusing; lines 285-287 and 290-294 are repeated in lines 305-307. Figure 6 is explained very chaotically (lines 349 - 354).
5) The performance results are competitive, but is the processing time or computational operations an advantage of this approach?
6) In another unlabeled dataset, how do authors define the outlier data?
Your model should be compared to other datasets to show the generality of the proposed approach.
Author Response
Comment 1: The abstract should contain less context on IoT and include this proposed method’s novelty and advantages
Author response: Thank you for your comment.
Author action: We have updated it and in red as follows
Internet of Things (IoT) refers to a system of interconnected, internet-connected devices and sensors that allows for collecting and dissemination of data. The data provided by these sensors may include outliers or exhibit anomalous behavior as a result of attack activities or device failure, for example. However, the majority of existing outlier detection algorithms rely on labeled data, which is frequently hard to obtain in IoT domain. More crucially, the IoT's data volume is continually increasing, necessitating the requirement for predicting and identifying the classes of future data. In this study, we propose an unsupervised technique based on deep Variational Auto-Encoder (VAE) to detect outliers in IoT data by leveraging the characteristic of the reconstruction ability and the low-dimensional representation of the input data's latent variables of the VAE. First, the input data is standardized. Then, we employ the VAE to find a reconstructed output representation from the low dimensional representation of the latent variables of the
input data. finally, the reconstruction error between the original observation and reconstructed one is used as outlier score. Our model was trained only using normal data with no labels in unsupervised manner and evaluated using Statlog (Landsat Satellite) dataset. The unsupervised model achieved a promising and comparable results with the state-of-the-art outlier detection schemes with precision ≈ 90% and F1-score = 79%.
Comment 2: Pictures of figure 2 show low resolution; this diagram should include the number of classes and more detail about the use of the dataset (80%-20%).
Author response: Thank you for your comment.
Author action: Regaring the use of the dataset (80%-20%), The figure was updated and it is also cleared and modified in both subsection 3.1 and section 4 as follows.
Subsection 3.1
Data is central to deep learning models, and it serves as fuel for these learning models. We gathered satellite dataset in tabular form for this project from a publicly ac-cessible data source. Statlog (Landsat Satellite) is a multi-class classification dataset from the UCI machine learning repository. The Statlog (Landsat Satellite) contains a total of 6435 samples where the total normal samples = 4399 and the total outlier samples = 2036. These samples are stored in tabular form and have 36 features or dimensions.
Section 4:
Only normal samples were used to train the model, so outlier samples from the training set were excluded and added to the test set. Hence, the model was tested using all 2036 outlier samples besides the test normal samples.
Comment 3: Deep VAE is theoretically few described; subsection 2.3 should be in section 3. Moreover, section 3 is confused; please check subsections 3.2 and 3.3.
Author response: Thank you for your comment.
Author action: We descriped VAE in more detailed, moved subsection 2.3 to be in section 3.3 and checked, modified and updated section 3 and all its subsections as follows:
3.3 Variational Auto-Encoder Algorithm
A variational autoencoder (VAE) is a type of generative models in which the distribution of the encodings is regularized during training. In other words, VAE give the latent space structure. The internal structure of the multidimensional latent space for a well-learned model defines its properties. The decoder component reconstructs the input using this information. The VAE architecture is made up of an encoder and a decoder with an intermediate compressed low-dimensional layer space, just as the AE architecture. The encoder maps the data to a posterior distribution, which is exclusive to VAE. Use of the univariate Gaussian distribution is a popular option in VAE. The Kullback-Leibler divergence measure between the approximation output and the target (input) characterizes the regularized term. The sum of the reconstruction loss and the Kullback-Leibler divergence is the overall loss in VAE as shown in Algorithm 1.
A VAE has been shown to be successful in outlier detection as well as providing a wide range of complicated data. This assumes that normal samples' latent representation is compatible with a Gaussian distribution. It means that all training datasets are grouped in feature space, and anomalies are located distant from the cluster. [47]. Through a latent space, VAE can acquire the feature distribution for a high-dimensional input data. The VAE model's framework is shown in Figure
Comment 4: Experimental results are also confusing; lines 285-287 and 290-294 are repeated in lines 305-307. Figure 6 is explained very chaotically (lines 349 - 354).
Author response: Thank you for your comment.
Author action: The duplicated lines were removed and updated.
Comment 5: The performance results are competitive, but is the processing time or computational operations an advantage of this approach?
Author response: Thank you for your comment.
Author action: As the GPUs and Processing units evolve rapidly, we focus in our work on the accuracy of the outlier detection and identification. It worths to mention that our techniques has promising processing time but it is not our main goal in this research.
Comment 6: In another unlabeled dataset, how do authors define the outlier data?
Author response: Thank you for your comment.
Author action:. We can't run tests without outlier samples because we operate unsupervised with only normal samples during training. As a result, in order to distinguish between normal and outliers, we must choose a threshold that needs the presence of both classes in the test set.
Comment 7: Your model should be compared to other datasets to show the generality of the proposed approach.
Author response: Thank you for your comment.
Author action: It is vital to note that comparing outcomes in this area of unsupervised learning is highly challenging owing to the wide range of accessible datasets and techniques. Furthermore, the clustering labels may seem different and the performance indicators may vary. As certain dataset labels are never completely recognized, it may result in presenting findings in ambiguous test datasets. It is evident from this that the performance of the predicted FC-VAE may change if the prediction is based on a different set of training or test datasets.
The proposed FC-VAE model was tested on the KPI time series dataset used fro VAE-GEN [54]. We made a public request to the researchers to reveal the dataset's specifics. The authors were gracious enough to publish their model's source code as well as the dataset on Jupiter Notebook. We executed FC-VAE model on their dataset and attained the promised accuracy of % [reference]. Table 3 compares the accuracy acquired as a consequence of this comparison experiment with the Statlog dataset used for FC-VAE. It is incredible to note that the satellite dataset provided considerably better findings in terms of precision and F1 score, indicating the generality of the proposed approach.
Table 2. Comparison of Proposed FC-VAE with state of the art techniques
|
Method |
Dataset |
SUP. |
Results |
|
VAE-GAN |
Time series |
UnSep |
P 0.76, R 0.5, F1 0.6 |
|
FC-VAE |
Statlog |
UnSup |
P 0.897, R 0.706, F1 0.79 |

Reviewer 2 Report
The article addresses the topical issue of data protection on the Internet, detection and preemption of attacks, unauthorized access, and other deviations in satellite data.The authors have made a comparison and comparative analysis of the results obtained when applying other similar methods presented in the scientific literature.
The reviewer thinks the article will be of interest to the journal's readers and suggests
that the editors accept it.
The reviewer has the following remarks and recommendations to the authors:
1. The methodology used should be described in more detail.
2. What exactly are the contributions/claims of the authors.
3. Does the study have any limitations?
4. What exactly do the anomaly detection tests consist of?
5. To provide the sources from which the proposed evaluation parameters were used.
6. Authors must complete: Author Contributions, Funding and Conflicts of Interest.
Author Response
Comment 1: The methodology used should be described in more detail.
Author response: Thank you for your comment.
Author action: We have updated it and tried to describe it in more details as possible in red.
Comment 2: What exactly are the contributions/claims of the authors.
Author response: Thank you for your comment.
Author action: The contribution demonstrates the benefit of using the VAE as an unsupervised model to find outliers in IOT data with decent detection accuracy.
Comment 3: Does the study have any limitations?
Author response: Thank you for your comment.
Author action: The only constraint is in the test set. We can't run tests without outlier samples because we operate unsupervised with only normal samples during training. As a result, in order to distinguish between normal and outliers, we must choose a threshold that needs the presence of both classes in the test set.
Comment 4: What exactly do the anomaly detection tests consist of?
Author response: Thank you for your comment.
Author action: In the test, we apply our proposed model to a set of normal and outlier samples. The trained model can then produce an outlier score for each sample, indicating whether the sample is normal or outlier.
Comment 5: To provide the sources from which the proposed evaluation parameters were used.
Author response: Thank you for your comment.
Author action: the evaluation parameter of relevant research work is added as follows
The proposed evaluation parameters has been taken from existing research [51,52]
Comment 6: Authors must complete: Author Contributions, Funding and Conflicts of Interest.
Author response: Thank you for your comment.
Author action: We have added the contributions, funding and conflict of interest subsections as follows
Author Contributions: Conceptualization, X.X. and Y.Y.; methodology, X.X.; software, X.X.; validation, X.X., Y.Y. and Z.Z.; formal analysis, X.X.; investigation, S.T.; resources, X.X.; data curation, X.X.; writing—original draft preparation, S.T.; writing—review and editing, S.T.; visualization, X.X.; supervision, X.X.; project administration, X.X.; funding acquisition, Y.Y. All authors have read and agreed to the published version of the manuscript.”
Funding: This research was funded by NAME OF FUNDER, grant number XXX” and “The APC was funded by XXX”.
Institutional Review Board Statement: Not Applicable
Informed Consent Statement: Not applicable.
Data Availability Statement: Not applicable.
Conflicts of Interest: The authors declare no conflict of interest

Round 2
Reviewer 1 Report
Thank you to the authors for providing an updated manuscript and following some of my concerns. I still have some suggestions about concerns 2, 3, and 7 of the first round:
2. Figure 2 has increased the resolution, but this only highlight the poor quality of the image more than before. I used the term “resolution” instead of saying the quality of the image; this was my fault in the first round. Could the authors improve the quality of figure 2? Some pixels around drawings can be perceived, and usually, design software helps to avoid these defects.
Also, the authors have omitted to include notable information about the network architecture. For instance, I had mentioned including the percentage of the dataset used for training and testing. Still, this kind of overall diagram usually includes additional key parameters during architecture design to provide more relevant information on the methodology proposed.
3. The authors have moved subsection 2.3 to section 3, as I had recommended, but this action was performed without transitioning this information into the new section. In the current version, figure 2 is calling for the first time in line 251, without significant relevance, then a profound description of this figure is provided some paragraphs after. Could the authors fix this abrupt change in the text? Finally, I consider that figure 2 should be in the section called for the first time (or close to).
7. The information provided in the paragraphs before Table 2 is incomplete. Also, there is two Table 2.
Author Response
Comment 1 Figure 2 has increased the resolution, but this only highlight the poor quality of the image more than before. I used the term “resolution” instead of saying the quality of the image; this was my fault in the first round. Could the authors improve the quality of figure 2? Some pixels around drawings can be perceived, and usually, design software helps to avoid these defects.
Also, the authors have omitted to include notable information about the network architecture. For instance, I had mentioned including the percentage of the dataset used for training and testing. Still, this kind of overall diagram usually includes additional key parameters during architecture design to provide more relevant information on the methodology proposed.
Author response: Thank you for your comment.
Author action: We have updated it as follows
Comment 2: The authors have moved subsection 2.3 to section 3, as I had recommended, but this action was performed without transitioning this information into the new section. In the current version, figure 2 is calling for the first time in line 251, without significant relevance, then a profound description of this figure is provided some paragraphs after. Could the authors fix this abrupt change in the text? Finally, I consider that figure 2 should be in the section called for the first time (or close to).
Author response: Thank you for your comment.
Author action: We have updated it in red as follows:
- Materials and Methods
In this section we present the dataset description used in this research work. Once dataset is acquired, preprocessing is conducted on tabular dataset of Satellite (Landsat Satellite) Then, the VAE is described in details to show its working mechanism besides its importance in our work. After that, the proposed framework is presented that is based on VAE. In this study, we used a Satellite input database to train a VAE model.
3.2 Dataset pre-processing
The dataset of Satellite (Landsat Satellite) was available in tabular form. We standardize features by removing the mean and scaling to unit variance. This process is considered as a pre-processing step in the proposed framework.
3.3 Variational Auto-Encoder Algorithm
A variational autoencoder (VAE) is a type of generative models in which the distribution of the encodings is regularized during training. In other words, VAE give the latent space structure. The internal structure of the multidimensional latent space for a well-learned model defines its properties. The decoder component reconstructs the input using this information. The VAE architecture is made up of an encoder and a decoder with an intermediate compressed low-dimensional layer space, just as the AE architecture. The encoder maps the data to a posterior distribution, which is exclusive to VAE. Use of the univariate Gaussian distribution is a popular option in VAE. The Kullback-Leibler divergence measure between the approximation output and the target (input) characterizes the regularized term. The sum of the reconstruction loss and the Kullback-Leibler divergence is the overall loss in VAE as shown in Algorithm 1.
A VAE has been shown to be successful in outlier detection as well as providing a wide range of complicated data. This assumes that normal samples' latent representation is compatible with a Gaussian distribution. It means that all training datasets are grouped in feature space, and anomalies are located distant from the cluster. [47]. Through a latent space, VAE can acquire the feature distribution for a high-dimensional input data.
This study employs a fully-connected neural network in both the encoder and decoder levels [48].
|
Algorithm 1 |
sVAE base outlier detector |
|
Input |
Normal dataset D |
|
|
|
|
|
Dataset with abnormal d (i)={1,2,3……N} |
|
|
A thresh hold value α |
|
Output |
Reconstruction probability class labels |
|
Steps |
Ð¤Ñ²ß train using VAE with normal dataset D |
|
1. |
For loop from k=1 to N |
|
2. |
Calculate
|
|
3. |
if Reconstruction error (k)>α then |
|
|
outlier ßd(k) |
|
|
Else |
|
|
Not outlier ß d(k) |
|
|
end if |
|
|
End for |
Where gθ(f∅(d^k)) is recognition network also known as probalitic encoder. f∅(d^k) is a generative decoder as generative network.
The VAE system provided here extracts useful characteristics for numerous standard unsupervised outlier detection applications. The reconstruction probability is computed using the stochastic latent variables that create the parameters from distribution of original input variable. The constraints of the input variable distribution, not the input variable itself, are reconstructed. This is the likelihood of data being generated from a particular latent variable pulled from the approximate posterior distribution.
Comment 3: The information provided in the paragraphs before Table 2 is incomplete. Also, there is two Table 2.
Author response: Thank you for your comment.
Author action: We have updated it in red as follows:
Table 3. Comparison of Proposed FC-VAE with state of the art techniques
|
Technique |
Dataset |
Learning Technique |
Results |
|
VAE-GAN |
Time series |
UnSup |
P 0.76, R 0.5, F1 0.6 |
|
FC-VAE |
Statlog |
UnSup |
P 0.897, R 0.706, F1 0.79 |
We have also added the missing details of accuracy in the paragraph.
